# TEACHING WITH COMMENTARIES

**Aniruddh Raghu** *
MIT
araghu@mit.edu

**Maithra Raghu**
Google Research

**Simon Kornblith**
Google Research

**David Duvenaud**
Google Research & University of Toronto

**Geoffrey Hinton**
Google Research & University of Toronto

## ABSTRACT

Effective training of deep neural networks can be challenging, and there remain many open questions on how to best learn these models. Recently developed methods to improve neural network training examine *teaching*: providing learned information during the training process to improve downstream model performance. In this paper, we take steps towards extending the scope of teaching. We propose a flexible teaching framework using *commentaries*, learned meta-information helpful for training on a particular task. We present gradient-based methods to learn commentaries, leveraging recent work on implicit differentiation for scalability. We explore diverse applications of commentaries, from weighting training examples, to parameterising label-dependent data augmentation policies, to representing attention masks that highlight salient image regions. We find that commentaries can improve training speed and/or performance, and provide insights about the dataset and training process. We also observe that commentaries *generalise*: they can be reused when training new models to obtain performance benefits, suggesting a use-case where commentaries are stored with a dataset and leveraged in future for improved model training.

## 1 INTRODUCTION

Training, regularising, and understanding complex neural network models is challenging. There remain central open questions on making training faster and more data-efficient (Kornblith et al., 2019; Raghu et al., 2019a;b), ensuring better generalisation (Zhang et al., 2016) and improving transparency and robustness (Bau et al., 2017; Madry et al., 2017). A promising approach for addressing these questions is *learning to teach* (Zhu, 2015), in which learned auxiliary information about a task is provided to a neural network to inform the training process and help downstream objectives. Examples include providing auxiliary training targets (Liu et al., 2019; Navon et al., 2020; Pham et al., 2020) and reweighting training examples to emphasise important datapoints (Fan et al., 2020; Jiang et al., 2018; Ren et al., 2018; Shu et al., 2019).

Learning to teach approaches have achieved promising results in vision and language applications (Jiang et al., 2018; Ren et al., 2018; Shu et al., 2019; Hu et al., 2019) using a handful of specific modifications to the training process. In this paper, we take steps towards generalising these approaches, introducing a flexible and effective learning to teach framework using *commentaries*. Commentaries represent learned meta-information helpful for training a model on a task, and once learned, such commentaries can be reused as is to improve the training of new models. We demonstrate that commentaries can be used for applications ranging from speeding up training to gaining insights into the neural network model. Specifically, our contributions are:

1. We formalise the notion of *commentaries*, providing a unified framework for learning meta-information that can be used to improve network training and examine model learning.
2. We present gradient-based methods to learn commentaries by optimising a network's validation loss, leveraging recent work in implicit differentiation to scale to larger models.
3. We use commentaries to define example-weighting curricula, a common method of teaching neural networks. We show that these learned commentaries hold interpretable insights, lead to speedups in training, and improve performance on few-shot learning tasks.

---

*Work done while interning at Google.

4. We define data augmentation policies with label-dependent commentaries, and obtain insights into the design of effective augmentation strategies and improved performance on benchmark tasks as compared to baselines.
5. We parameterise commentaries as attention masks to find important regions of images. Through qualitative and quantitative evaluation, we show these masks identify salient image regions and can be used to improve the robustness of neural networks to spurious background correlations.
6. We show that learned commentaries can *generalise*: when training new models, reusing learned commentaries can lead to learning speed/performance improvements. This suggests a use-case for commentaries: being stored with a dataset and leveraged to improve training of new models.

## 2 TEACHING WITH COMMENTARIES

**Definition:** We define a *commentary* to be learned information helpful for (i) training a model on a task or (ii) providing insights on the learning process. We envision that commentaries, once learned, could be stored alongside a dataset and reused as is to assist in the training of new models. Appendix A explores a simple instantiation of commentaries for Celeb-A (Liu et al., 2015), to provide intuition of the structures that commentaries can encode.

Formally, let $t(x, y, i; \phi)$ denote a commentary that is a function of a data point $x$, prediction target $y$, and iteration of training $i$, with parameters $\phi$. The commentary may be represented in a tabular fashion for every combination of input arguments, or using a neural network that takes these arguments as inputs. The commentary is used to train a student network $n(x; \theta)$ with parameters $\theta$.

### 2.1 LEARNING COMMENTARIES

We now describe algorithms to learn commentaries [1]. Throughout, we denote the training set as $\mathcal{D}_T$, the validation set as $\mathcal{D}_V$ and the loss function (e.g. cross-entropy) as $\mathcal{L}$. With $\theta$ denoting the parameters of the student network and $\phi$ denoting the commentary parameters, we let $\hat{\theta}$, $\hat{\phi}$ be the respective optimised parameters. We seek to find $\hat{\phi}$ such that the student network's validation loss, $\mathcal{L}_V$, is minimised. As the commentary is used during the training of the student network, $\mathcal{L}_V$ implicitly depends on $\phi$, enabling the use of gradient-based optimisation algorithms to find $\hat{\phi}$.

**Algorithm 1: Backpropagation Through Training:** When student network training has a small memory footprint, we optimise commentary parameters by iterating the following process, detailed in Algorithm 1: (1) train a student and store the computation graph during training; (2) compute the student's validation loss; (3) calculate the gradient of this loss w.r.t. the commentary parameters by backpropagating through training; (4) update commentary parameters using gradient descent.

By optimizing the commentary parameters over the entire trajectory of student learning, we encourage this commentary to be effective when used in the training of new student networks. This supports the goal of the commentary being stored with the dataset and reused in future model learning.

**Algorithm 2: Large-Scale Commentary Learning with Implicit Differentiation:** When training the student model has a large memory footprint, backpropagating through training to obtain exact commentary parameter gradients is too memory expensive. We therefore leverage the Implicit Function Theorem (IFT) and efficient inverse Hessian approximation to obtain approximate gradients, following Lorraine et al. (2020).

The gradient of the validation loss w.r.t. the commentary parameters can be expressed as:

$$\frac{\partial \mathcal{L}_V}{\partial \phi} = \frac{\partial \mathcal{L}_V}{\partial \hat{\theta}} \times \frac{\partial \hat{\theta}}{\partial \phi}. \tag{3}$$

The first term on the right hand side in equation 3 is simple to compute, but the second term is expensive. Under fixed-point and regularity assumptions on student and commentary parameters $(\hat{\theta}(\phi), \phi)$, the IFT allows expressing this second term $\frac{\partial \hat{\theta}}{\partial \phi}$ as the following product:

$$\frac{\partial \hat{\theta}}{\partial \phi} = - \left[ \frac{\partial^2 \mathcal{L}_T}{\partial \theta \, \partial \theta^T} \right]^{-1} \times \frac{\partial^2 \mathcal{L}_T}{\partial \theta \, \partial \phi^T} \Big|_{\hat{\theta}(\phi)}, \tag{4}$$

---

[1]Code at https://github.com/googleinterns/commentaries

---

**Algorithm 1** Commentary Learning through Backpropagation Through Training.

---

1: Initialise commentary parameters $\phi$
2: **for** $t = 1, \ldots, T$ meta-training steps **do**
3:     Initialise student network $n(x; \theta)$ with parameters $\theta_0$
4:     Train student network with $N$ steps of gradient descent to optimise:

$$\mathcal{L}_T(\theta, \phi) = \mathbb{E}_{x,y \sim \mathcal{D}_T} \left[ \tilde{\mathcal{L}} \left( n\left(x; \theta\right), t\left(\cdot\,; \phi\right), y \right) \right], \tag{1}$$

    where $\tilde{L}$ is a loss function adjusted from $\mathcal{L}$ to incorporate the commentary, and $\mathcal{L}_T(\theta, \phi)$ is the expected adjusted loss over the training data. Output: $\hat{\theta}$, the optimised parameters of student network (implicitly a function of $\phi$, $\hat{\theta}(\phi)$).
5:     Compute validation loss:

$$\mathcal{L}_V(\phi) = \mathbb{E}_{x,y \sim \mathcal{D}_V} \left[ \mathcal{L}\left( n(x; \hat{\theta}\,(\phi)), y \right) \right] \tag{2}$$

6:     Compute $\frac{\partial \mathcal{L}_V(\phi)}{\partial \phi}$, by backpropagating through the $N$ steps of student training, and update $\phi$.
7: **end for**
8: Output: $\hat{\phi}$, the optimised parameters of the commentary.

---

**Algorithm 2** Commentary Learning through Implicit Differentiation.

---

1: Initialise commentary parameters $\phi$ and student network parameters $\theta$
2: **for** $t = 1, \ldots, M$ **do**
3:     Compute the student network's training loss, $\mathcal{L}_T(\theta, \phi)$, equation 1.
4:     Compute the gradient of this loss w.r.t the student parameters $\theta$.
5:     Perform a single gradient descent update on the parameters to obtain $\hat{\theta}$ (implicitly a function of $\phi$, $\hat{\theta}(\phi)$).
6:     Compute the student network's validation loss, $\mathcal{L}_V(\phi)$, equation 2.
7:     Compute $\frac{\partial \mathcal{L}_V}{\partial \hat{\theta}}$.
8:     Approximately compute $\frac{\partial \hat{\theta}}{\partial \phi}$ with equation 4, using a truncated Neumann series with a single term and implicit vector-Jacobian products (Lorraine et al., 2020).
9:     Compute the overall derivative $\frac{\partial \mathcal{L}_V}{\partial \phi}$ using steps (7) and (8), and update $\phi$.
10:    Set $\theta \leftarrow \hat{\theta}$.
11: **end for**
12: Output: $\hat{\phi}$, the optimised parameters of the commentary.

---

i.e., a product of an inverse Hessian and a matrix of mixed partial derivatives. Following Lorraine et al. (2020), we efficiently approximate this product using a truncated Neumann series and implicit vector-Jacobian products. Leveraging this approximation then yields a second method for commentary learning, described in Algorithm 2. Since a single term in the Neumann series is sufficient for learning, each iteration of this algorithm has similar time complexity to a single iteration of training.

In this method, commentary parameters are learned jointly with student parameters, avoiding training a single student model multiple times. This approach therefore scales to millions of commentary parameters and large student models (Hataya et al., 2020; Lorraine et al., 2020). However, since the commentary is not directly optimised over the entire trajectory of learning, its generalisability to new models is not ensured. We examine this in our experiments, demonstrating that commentaries learned in this manner can indeed generalise to training new student networks.

## 3   COMMENTARIES FOR EXAMPLE WEIGHTING CURRICULA

We now explore our first main application of commentaries: encoding a separate weight for each training example at each training iteration. Since the commentaries are a function of the training iteration, they can encode curriculum structure, so we refer to them as curriculum commentaries.

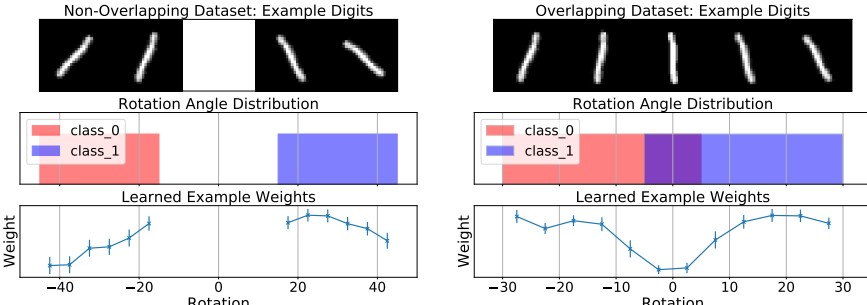

**Figure 1: Learned per-example training weights on Rotated MNIST.** We visualise rotated digits and rotation distributions from both the datasets. We plot the learned example weights at iteration 500 of student training as a function of rotation after discretising based on digit rotation. In the non-overlapping case (left), examples closer to the decision boundary are weighted most. When classes overlap (right), the more representative examples with greater rotation are weighted highly and examples in the overlap region are downweighted, which is sensible as examples in the overlap region are less indicative of the class label.

We specify these weights using a *commentary neural network (or teacher network)* $t(x, i; \phi) \rightarrow [0, 1]$ that produces a weight for every training example at every iteration of training of the student network. When training a student network, using the notation of §2.1, the commentary is incorporated in the training loss as: $\tilde{\mathcal{L}} = t(x, i; \phi) \cdot \mathcal{L}\big(n(x; \theta), y\big)$, where $\mathcal{L}(\cdot)$ is the original loss function for the task. The validation loss is unweighted.

### 3.1 SYNTHETIC EXAMPLE: ROTATED MNIST DIGITS

We first learn example weight curriculum commentaries on a synthetic MNIST binary classification problem. Each example in the dataset is a rotated MNIST digit '1', with variable rotation angle that defines the class. We generate two datasets: the **non-overlapping dataset** and the **overlapping dataset**. In the non-overlapping dataset, the rotation angle for each example from class 1 and class 0 is drawn from non-overlapping distributions Uniform$[15, 45]$ and Uniform$[-45, -15]$ respectively. In the overlapping dataset, the rotation angles are drawn from overlapping distributions Uniform$[-5, 30]$ and Uniform$[-30, 5]$ respectively (Figure 1).

We use two block CNNs as both the commentary neural network and student network. The commentary network takes as input the image and the iteration of student training, and outputs a weight for each example in the batch. We learn commentary parameters by backpropagating through student training (Algorithm 1, §2.1), and use 500 gradient steps for inner optimisation (i.e., $N = 500$). Implementation is with the `higher` library (Grefenstette et al., 2019). Further details in Appendix B.1.

**Results:** Figure 1 visualises the two datasets and plots the learned example weights as a function of rotation at iteration 500 of the student training. When classes do not overlap (left), the example weights are highest for those examples near to the decision boundary (small rotation magnitude). When the classes do overlap (right), the more representative examples further from the boundary are upweighted and ambiguous examples in the overlap region are downweighted: a sensible result. We perform further analysis of the learned example weighting curriculum in Appendix B.1, demonstrating that the learned curricula in both cases are meaningful. Overall, these results demonstrate that the learned commentaries capture interesting and intuitive structure.

### 3.2 COMMENTARIES FOR CIFAR10 AND CIFAR100

We now learn example weighting curriculum commentaries on CIFAR10 and CIFAR100. The commentary network is again the two block CNN architecture, and when training the commentary network, the student network is also a two block CNN. We use Algorithm 1, §2.1 once more, with 1500 gradient steps in the inner optimisation: $N = 1500$. For evaluation, the trained commentary network is used to produce example weights for (i) 2 block CNN (ii) ResNet-18 (iii) ResNet-34 student networks, all trained for 25000 steps, considering 3 random initialisations, to assess generalisability. Further details in Appendix B.2.

Commentary Generalisation Over Training Steps

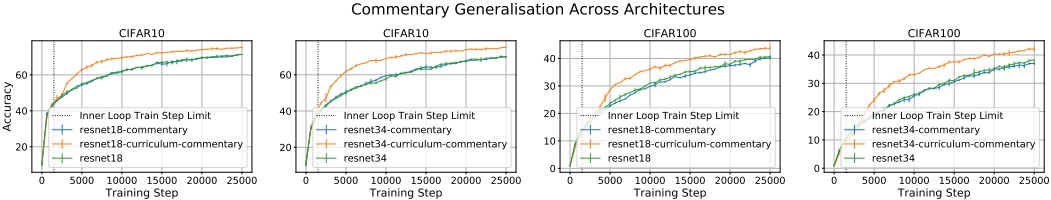

**Figure 2: Example weight curricula can speed up training.** Test-set accuracy curves on CIFAR10/100 when using curriculum commentaries, non-curriculum commentaries, and no commentaries, during student network training. The learned curriculum commentary network which generates per-iteration example weights results in learning speed improvements. This learning speed improvement holds when the student network is trained for many more steps than the number of inner loop update steps used during commentary network training (1500 steps). This demonstrates that the commentaries generalise to longer training times.

Commentary Generalisation Across Architectures

**Figure 3: Example weight curricula generalise across network architectures.** Using a learned curriculum commentary network trained with a simple 2 block CNN student, we apply these example weights to train two ResNet architectures. This gives improved test set accuracy curves for ResNet students also, indicating that the commentaries generalise across architectures.

**Example weighting commentaries improve learning speed.** Figure 2 shows accuracy curves on the test sets of CIFAR10/100 for the two block CNN student with example weight curricula (orange line), a baseline (green line, no example weights) and an ablation (blue line, example weights without curriculum structure, meaning the commentary network only takes the image $x$ and not the training iteration $i$ as an argument when outputting weights). On both datasets, the networks trained using the curriculum commentaries obtain better performance than the baseline and ablation over approximately 25000 steps of training (10 epochs), and have superior learning curves.

**Example weighting commentaries generalise to longer training times and across architectures.** At training time, the commentary network was learned to produce example weights for the two block CNN student for 1500 inner update steps ($N = 1500$, Algorithm 1). Figure 2 shows that the learned example weights lead to student network learning speedups well-beyond this point, suggesting generalisability of the commentaries to longer training times. In addition, when the same commentary network is used to produce example weights for ResNet-18/34 students (Figure 3), we also observe a learning speedup, suggesting that the commentaries can generalise across architectures.

### 3.3  COMMENTARIES FOR FEW-SHOT LEARNING

Finally, we use example weight commentaries for few-shot learning. We start with the MAML algorithm (Finn et al., 2017), which learns a student parameter initialisation that allows fast adaptation on a new learning task using a support set of examples for that task. To incorporate example weighting in MAML, at training time, we jointly learn the MAML initialisation and a commentary network to provide per-example weights for each example in the support set, as a function of the inner loop step number. At test time, we follow the standard MAML procedure, and also incorporate the example weights when computing the support set loss and the resulting updates. Both networks use the 4-conv backbone structure from the original MAML paper. Details are in Appendix B.3.

We evaluate a standard MAML baseline and our commentary variant on standard few-shot learning benchmarks: (i) training/testing on MiniImageNet (MIN); and (ii) training on MIN and testing on CUB-200-2011 (CUB). Results are shown in Table 1, specifying the experimental setting ($N$-way

| Mode: Train Data → Test Data | MAML Baseline | Example weighting + MAML |
|---|---|---|
| 5-way 1-shot: MIN→MIN | $43.64 \pm 0.61$ | $45.07 \pm 0.61$ |
| 5-way 1-shot: MIN→CUB | $37.45 \pm 0.55$ | $38.00 \pm 0.54$ |
| 5-way 5-shot: MIN→MIN | $61.85 \pm 0.52$ | $62.99 \pm 0.52$ |
| 5-way 5-shot: MIN→CUB | $57.75 \pm 0.54$ | $59.06 \pm 0.55$ |

**Table 1:** Mean accuracy and 95% confidence interval across 1000 test-time tasks. On common few-shot learning benchmarks, using example weighting for the support set improves the MAML baseline's accuracy. Improvements are also observed when evaluating on out-of-distribution datasets.

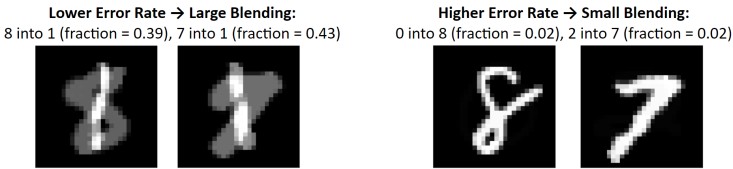

**Figure 4: The learned blending augmentation is related to the class error rate.** On digit 1s, a class with low average error rate, the learned augmentation blends in large amounts of other digits so as to maximise the learning signal. On digits with higher average error rates, such as 7s and 8s, the level of blending is very low so as to focus on classifying the class alone (rather than making the task more complex by adding another digit).

$K$-shot), and the dataset used for training/testing. In all experiments, incorporating example weighting can improve on the MAML baseline, suggesting the utility of these commentaries in few-shot learning. Further experiments on other benchmark datasets (CIFAR-FS/SVHN) showing similar trends are in the appendix (Table B.1).

## 4 COMMENTARIES FOR DATA AUGMENTATION

We now investigate label-dependent commentaries that parameterise data augmentation policies. We consider an augmentation scheme where pairs of images are blended together with a proportion dependent on the classes of the two examples. At each training iteration, we:

- Sample two examples and their labels, $(x_1, y_1)$ and $(x_2, y_2)$ from the training set.
- Obtain the blending proportion $\lambda = t(y_1, y_2; \phi)$, and form a new image $x_m = \lambda x_1 + (1 - \lambda)x_2$, and class $y_m$ equivalently.
- Use this blended example-label pair $(x_m, y_m)$ when computing the training loss.

To compute the validation loss, use only unblended examples from the validation set.

For classification problems with $N$ classes, the teacher $t(y_1, y_2; \phi)$ outputs an $N \times N$ matrix. This augmentation scheme is inspired by *mixup* (Zhang et al., 2018). However, we blend with a deterministic proportion, depending on pairs of labels, rather than drawing a blending factor from a Beta distribution. In doing so, we more finely control the augmentation policy.

**Augmentation Commentaries on MNIST:** We learn an augmentation commentary model $t$ on MNIST by direct backpropagating through the training of a 2-block CNN student network (Algorithm 1, §2.1). In the learned augmentation, the error rate on class $i$ is correlated (Pearson correlation= $-0.54$) with the degree of blending of other digits into an example of class $i$: lower error on class $i$ implies that other digits are blended more heavily into it. On MNIST, this means that the class that has on average the lowest error rate (class 1) has other digits blended into it significantly (Figure 4 left); classes that have on average higher error rate (e.g., class 7, class 8) rarely have other digits blended in (Figure 4 right). Further details in Appendix C.1.

### 4.1 AUGMENTATION COMMENTARIES FOR CIFAR10 AND CIFAR100

We next learn and evaluate augmentation commentaries on CIFAR10 and CIFAR100. We evaluate the effect of these augmentation commentaries on improving generalisation performance for a standard student network architecture. Since this is a memory intensive process, we use the implicit differentiation method (Algorithm 2, § 2.1) to ensure computational tractability. We learn the commentaries jointly with a ResNet-18 student network. Once the commentary is learned, we fix it and

|  | CIFAR10 | | CIFAR100 | |
|---|---|---|---|---|
|  | Test Accuracy | Test Loss | Test Accuracy | Test Loss |
| No commentary | $93.84 \pm 0.22$ | $0.239 \pm 0.005$ | $74.79 \pm 0.39$ | $1.05 \pm 0.03$ |
| Random commentary | $93.84 \pm 0.30$ | $0.385 \pm 0.007$ | $73.39 \pm 0.58$ | $1.14 \pm 0.01$ |
| *mixup* (Zhang et al., 2018) | $94.42 \pm 0.12$ | $0.313 \pm 0.010$ | $75.89 \pm 0.55$ | $1.01 \pm 0.02$ |
| Learned commentary | $94.12 \pm 0.19$ | $0.225 \pm 0.007$ | $76.25 \pm 0.11$ | $0.97 \pm 0.01$ |

**Table 2: Learned augmentation commentaries result in competitive or improved model accuracy and loss on CIFAR10/100.** Using the learned, label-dependent augmentation commentary during training results in models that are competitive with *mixup* and superior to other baselines. We show mean/standard deviation over three different initialisations.

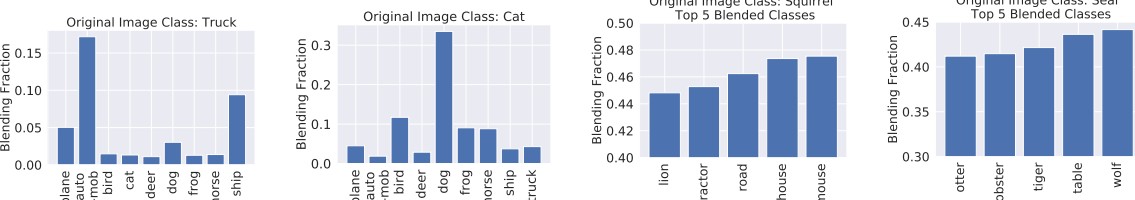

**Figure 5: Blending proportions for selected CIFAR10 and CIFAR100 classes reveal interesting insights.** Visualizing the learned blending proportions on two CIFAR10 classes (left), we see that both classes are most blended with others that are visually similar (truck-automobile, and cat-dog), which may help the network differentiate between them. On CIFAR100 (right), considering the top 5 blended classes in two cases, we observe again the presence of visually similar classes that may be confused (seal-otter, and squirrel-mouse), but also visually unrelated classes. These may provide extra learning signal with each example.

train three new students with different random initialisations to evaluate the commentary's efficacy, assessing the commentary's generalisability.

**Results:** Table 2 shows test accuracy for different augmentation policies on CIFAR10 and 100. We compare the learned commentary to using only standard data augmentations for CIFAR10/100 (No commentary) and a random initialisation for the commentary matrix (Random commentary). We also compare to *mixup* (Zhang et al., 2018). We observe that the learned commentary is competitive with *mixup* and improves on other baselines across both tasks. In the appendix, we compare to an ablation that destroys the structure of the learned commentary grid by shuffling it, and find that the unshuffled commentary does better (Table C.1).

**Further Analysis:** In Figure 5, we visualise: (i) for two CIFAR10 classes, the blending fractions (defined as $(1 - \lambda)$) associated with the other classes (left); and (ii) for two CIFAR100 classes, the blending fractions associated with the five most highly blended classes. For CIFAR10, we see that other classes that are visually similar and therefore sources of misclassification are blended more significantly. On CIFAR100, we see that within the top 5 blended classes, there are classes that are visually similar, but there are also blended classes that have far less visual similarity, which could be blended in to obtain more learning signal per-example. Further analysis in Appendix C.2.

## 5 ATTENTION MASK COMMENTARIES FOR INSIGHTS AND ROBUSTNESS

We study whether commentaries can learn to identify key (robust) features in the data, formalising this problem as one of learning commentaries which act as attention masks. We learn commentary attention masks on a variety of image datasets: an MNIST variant, CIFAR10/100, medical chest X-rays, and Caltech-UCSD Birds (CUB)-200-2011, where we find that the learned commentaries identify salient image regions. Quantitatively, we demonstrate the effectiveness of attention mask commentaries over baselines in ensuring robustness to spurious correlations.

Formally, we learn a commentary network $t(x; \phi) \rightarrow [i, j]$ to output the centre of a 2D Gaussian that is then computed and used (with predefined standard deviation depending on the input image size, see Appendix D) as a pixelwise mask for the input image before feeding it through a student network. We denote the mask based on $t(x; \phi)$ as $m(x, t)$. Our goal is to learn masks that highlight the most

**Figure 6: Learned attention masks highlight salient image regions for classification.** We learn a commentary network to produce image attention masks on several image datasets, and the masks are qualitatively sensible in all cases. On Coloured MNIST, where the image label is determined by the red digit, the masks focus on the red digit. On a dataset of chest X-rays, the masks focus on the chest cavity, which is the appropriate reason for detecting the condition in question (cardiomegaly). On CIFAR10/100, the masks are focused on important regions, such as the faces of animals, the hump of the camel, and the bodies of vehicles.

important regions of the image for training and testing, so the masks are used both at train time and test time. We therefore have that $\tilde{\mathcal{L}} = \mathcal{L} = \text{x-ent}\left(n\left(x \odot m\left(x, t\right); \theta\right), y\right)$. The commentary network is a U-Net (Ronneberger et al., 2015) with an output layer from KeypointNet (Suwajanakorn et al., 2018). Commentary parameters are learned using Algorithm 2, §2.1, for a ResNet-18 student.

### 5.1 QUALITATIVE AND QUANTITATIVE ANALYSIS ON IMAGE DATASETS

**Masks for Coloured MNIST:** We learn masks on a dataset where each image has two MNIST digits, coloured red and blue, with the red digit determining the label of the image. As seen in Figure 6 left, the commentary selectively attends to the red digit and not the blue digit.

**Masks for Chest X-rays:** Using a dataset of chest X rays (Irvin et al., 2019), we train a student network to detect cardiomegaly, a condition where an individual's heart is enlarged. Learned masks are centered on the chest cavity, around the location of the heart (Figure 6), which is a clinically relevant region. These masks could be used in medical imaging problems to prevent models relying on spurious background features (Badgeley et al., 2019; Winkler et al., 2019).

**Masks for CIFAR10/100:** The learned masks on CIFAR10/100 (Figure 6) attend to important image regions that define the class, such as the faces of animals/the baby, wheels/body of vehicles, and the humps of the camel. In the appendix (Table D.1) we show quantitatively that the learned masks are superior to other masking baselines, and also provide further qualitative examples.

### 5.2 MASK COMMENTARIES FOR ROBUSTNESS TO BACKGROUND CORRELATIONS

Using the task introduced in Koh et al. (2020), we now demonstrate that mask commentaries can provide robustness to spurious background correlations. We take the CUB-200-2011 dataset (Welinder et al., 2010) and the associated fine-grained classification problem to classify the image as one of 200 bird species. Using the provided segmentation masks with the dataset, the background of each image is replaced with a background from the Places dataset (Zhou et al., 2017). For the training and validation sets, there is a specific mapping between the CUB class and the Places class for the background, but for the testing set, this mapping is permuted, so that the background features are now spuriously correlated with the actual image class.

We first learn an attention mask commentary network, and for evaluation, use this learned network when training a new ResNet-18 student (pretrained on ImageNet, as in Koh et al. (2020)). We assess student performance on the validation and test sets (with the same and different background mappings as the training set, respectively), considering three random seeds.

**Results:** Learned masks are shown in Figure 7; the masks are mostly focused on the bird in the image. In a quantitative evaluation (Table 3), we see that using the masks helps model performance significantly on the test set as compared to a baseline that does not use masks. This suggests that the masks are indeed helping networks to rely on more robust features in the images. The validation accuracy drop is expected since the model input is a limited region of the image.

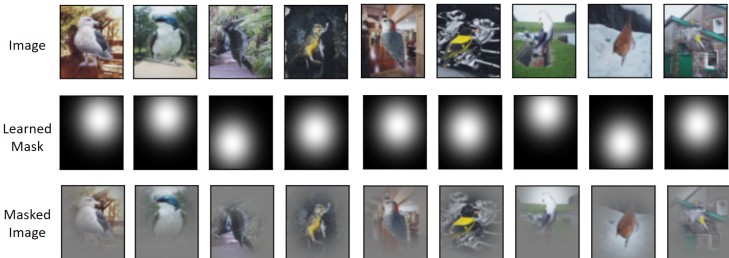

**Figure 7:** Learned masks on CUB dataset are primarily focused on the bird, rather than the background.

| | Validation Set Accuracy | | Test Set Accuracy (Distribution Shift) | |
| | Top 1 | Top 5 | Top 1 | Top 5 |
|---|---|---|---|---|
| Baseline (No masks) | $78.82 \pm 1.19$ | $93.95 \pm 0.28$ | $25.81 \pm 0.60$ | $56.98 \pm 0.69$ |
| With Masking Network | $74.73 \pm 0.76$ | $92.45 \pm 0.23$ | $30.46 \pm 0.27$ | $61.57 \pm 0.70$ |

**Table 3: Learned attention masks offer improved robustness to spurious background correlation.** We train a commentary network to produce attention masks on a version of the CUB-200-2011 dataset that contains spurious background correlations, with the correlations present in the training/validation set *not* present in the test set (the test set has a distribution shift). On both top 1 and top 5 accuracy, using the masks results in improved model performance on the shifted test set.

## 6 RELATED WORK

**Learning to Teach:** Neural network teaching and curriculum learning have been proposed as early as Bengio et al. (2009); Zhu (2015). Recent work examining teaching includes approaches to: select, weight, and manipulate training examples (Fan et al., 2018; Liu et al., 2017; Fan et al., 2020; Jiang et al., 2018; Ren et al., 2018; Shu et al., 2019; Hu et al., 2019); adjust loss functions (Wu et al., 2018); and learn auxiliary tasks/labels for use in training (Liu et al., 2019; Navon et al., 2020; Pham et al., 2020). In contrast to most of these methods, our work on commentaries aims to unify several related approaches and serve as a general learning framework for teaching. This enables applications in both standard settings such as example weighting, and also novel use-cases (beyond model performance) such as attention masks for interpretability and robustness. These diverse applications also provide insights into the training process of neural networks. Furthermore, unlike many earlier works that jointly learn teacher and student models, we also consider a different use case for learned commentaries: instead of being used in the training of a single model alone, we demonstrate that commentaries can be stored with a dataset and reused when training new models.

**Learning with Hypergradients:** Our algorithm for learning commentaries uses *hypergradients* — derivatives of a model's validation loss with respect to training hyperparameters. Prior work has proposed different approaches to compute hypergradients, including memory-efficient exact computation in Maclaurin et al. (2015), and approximate computation in Lorraine et al. (2020); Lorraine and Duvenaud (2018); MacKay et al. (2019). We build on Lorraine et al. (2020), which utilises the implicit function theorem and approximate Hessian matrix inversion for efficiency, to scale commentary learning to larger models.

## 7 CONCLUSION

In this paper, we considered a general framing for teaching neural networks using *commentaries*, defined as meta-information learned from the dataset/task. We described two gradient-based methods to learn commentaries and three methods of applying commentaries to assist learning: example weight curricula, data augmentation, and attention masks. Empirically, we show that the commentaries can provide insights and result in improved learning speed and/or performance on a variety of datasets. In addition, we demonstrate that once learned, these commentaries can be reused to improve the training of new models. Teaching with commentaries is a proof-of-concept idea, and we hope that this work will motivate larger-scale applications of commentaries and inspire ways of automatically re-using training insights across tasks and datasets.

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

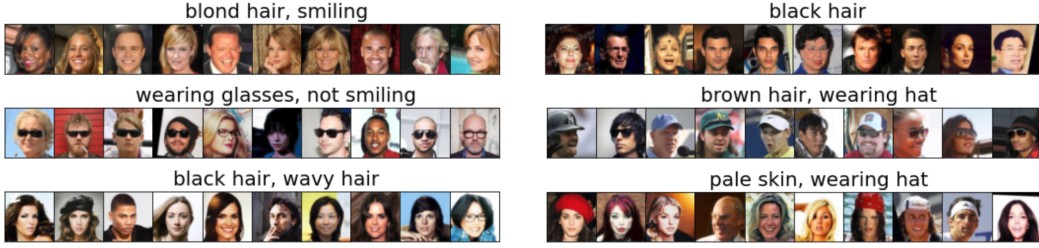

**Figure A.1: Dimensions of the metalearned commentaries show correlations with CelebA attributes.**

# A    ILLUSTRATIVE EXAMPLE: VECTOR COMMENTARIES ON CELEB-A

As an illustrative example, we describe how to learn real vector valued commentaries on the CelebA dataset, and then analyse the resulting commentaries.

The CelebA dataset contains images of faces, with each image also accompanied by 40 dimensional attribute labels. We learn a commentary neural network $t(x; \phi) \to [-1, 1]^{40}$ to produce a real vector valued commentary for each training example. We train a student network $n(x; \theta) \to (\hat{y}, \hat{t})$ to predict both the attributes and regress the commentary. The training and validaiton losses are as follows:

$$\tilde{\mathcal{L}} = \left(\hat{t} - t(x; \phi)\right)^2 + \text{x-ent}\big(n(x; \theta), y\big), \tag{5}$$

$$\mathcal{L} = \text{x-ent}\big(n(x; \theta), y, \big). \tag{6}$$

For both commentary and student networks, we use a 4-block CNN with $3 \times 3$ filters (similar to the CNN architectures from (Finn et al., 2017)). We learn commentary parameters using backprop-agation through student training (Algorithm 1, §2.1). The inner optimisation for student learning consists of 100 iterations of Adam optimzer updates on the training loss using a batch size of 4. We use the `higher` library (Grefenstette et al., 2019). We use 50 meta-iterations (outer optimisation iterations) to train the commentary network.

**Results:** Figure A.1 visualises the examples that are most positively/negatively correlated with certain dimensions of the vector valued commentaries. We see clear correspondence between interpretable attributes and the commentary dimensions. Attributes such as hair colour, wearing glasses, and wearing hats are well-encoded by the commentaries. This suggests that such commentaries could function as an auxiliary task to give a network more signal during the learning process. Storing the commentary network with the dataset could then allow these auxiliary labels to be re-used when training future models.

# B    EXAMPLE WEIGHTING CURRICULA

We provide further details about the experiments using commentaries to define example weighting curricula.

## B.1    ROTATED MNIST

**Dataset:** Both the overlapping and non-overlapping datasets are generated to have 10000 training examples, 5000 validation examples, and 5000 test examples.

**Network architectures:** CNNs with two blocks of convolution, batch normalisation, ReLU, and max pooling are used for both the student network and commentary network. The commentary network additionally takes in the iteration of student training, which is normalised and passed in as an extra channel in the input image (i.e., the MNIST image has two channels, instead of a single channel). The commentary network uses sigmoid activation to produce an example weight for every data point in the batch.

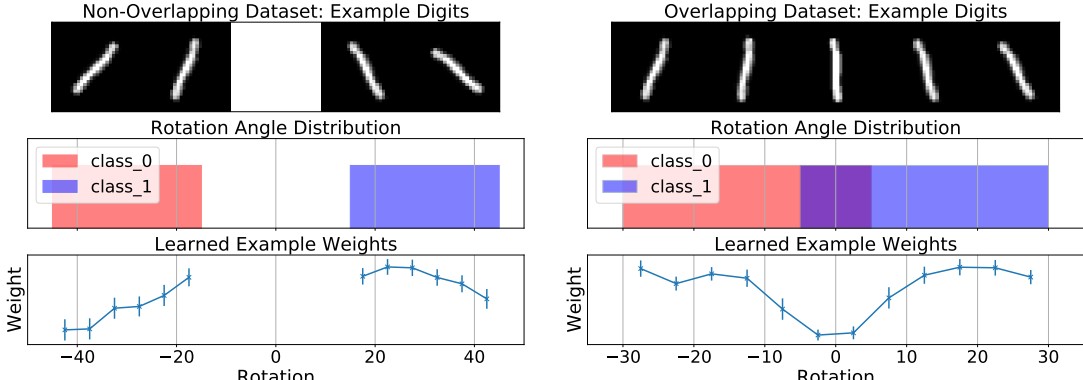

**Figure B.1: Learned per-example training weights on Rotated MNIST.** Visualizing the relationship between the example weights and rotations at the end of training (left) reveals that for the non-overlapping dataset, examples with the largest weights are located close to the decision boundary, which is expected, as these provide the most learning signal. In the overlapping case, weights for examples within the overlap region are lower, again expected as these examples are ambiguous and do not assist in learning.

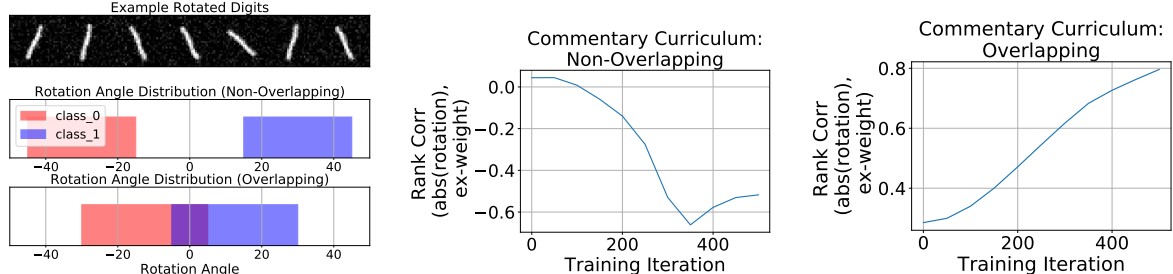

**Figure B.2: A learned curriculum of per-example training weights.** Example rotated digits and the two classes are shown on the left. We plot the rank correlation between the magnitude of digit rotation and the example weight over the course of student training for both datasets. When classes are non-overlapping, the curriculum weights examples which are closer to the decision boundary more as training goes on. This resembles teaching the student to focus on easy examples at first, and then moving to progressively harder examples as training progresses. When classes overlap, the curriculum weights more representative examples (near class centroids) more highly as training progresses.

**Training details:** We train both networks using the Adam optimiser, with a learning rate of 1e-4 for the student, and 1e-3 for the commentary network. The student network is trained for 500 inner optimisation steps, with a batch size of 10. We train for 20 commentary network iterations. Training is implemented using the `higher` library (Grefenstette et al., 2019).

**Results:** Figure B.1 visualises for both datasets: the relation between the learned example weight and the rotation of the digit at the end of student training (iteration 500) following binning. Considering these final example weights, for the non-overlapping case, examples with higher weight are those with smaller rotation and thus closer to the decision boundary (i.e., resembling **support vectors**) – these provide the most information to separate the classes. Lower weighted examples are those with greater rotation. When classes overlap, the weights are maximised for examples that better characterize each class, and are not in the overlap region – these are most informative of the class label.

Now considering the learned curriculum (Figure B.2), for the non-overlapping dataset, we observe that the rank correlation between the example weight and the rotation magnitude decreases over the course of student training iteration. This is intuitively sensible, since the example weights first prioritise the easy examples (large rotation), and then learn to focus on the harder ones (small rotation) that provide most information to separate the classes. By contrast, in the overlapping case, the

curriculum has examples that are further from the boundary (larger rotation magnitude) consistently weighted most highly, which is sensible as these are the most informative of the class label.

Overall, the results on this synthetic case demonstrate that the learned commentaries capture interesting and intuitive structure.

## B.2 CIFAR10/100

**Network architectures:** We use the two block CNN from the MNIST experiments for the commentary network, and as the student network when training the commentary network. We employ the same strategy for encoding the iteration of training. At testing time, we evaluate this commentary network by teaching three different student network architectures: two block CNN, ResNet-18, and ResNet-34.

**Training details:** We train both networks using the Adam optimiser, with a learning rate of 1e-4 for the student, and 1e-3 for the commentary network. During commentary network learning, the student network is trained for 1500 inner optimisation steps, with a batch size of 8, and is reset to random intialisation after each commentary network gradient step. We train for 100 commentary network iterations. Training is implemented using the `higher` library (Grefenstette et al., 2019). At testing time, we use a batch size of 64; the small batch size at training time is due to GPU memory constraints.

## B.3 FEW-SHOT LEARNING (FSL)

**Setup:** The MAML algorithm finds a network parameter initialisation that can then be used for adaptation to new tasks using a support set of examples. The commentary network here is trained to provide example weights for each example in the support set, at each iteration of inner loop adaptation (i.e., the example weights are not used at meta-testing time). We jointly learn the MAML initialisation and the commentary network parameters; intuitively, this represents learning an initialisation that is able to adapt to new tasks given examples and associated weights.

**Dataset details:** We use standard datasets used to evaluate FSL methods, and the associated splits between training and testing tasks from prior work (Lee et al., 2019; Long, 2018). We evaluate on two out-of-distribution settings, namely: training the few-shot learner on CIFAR-FS and testing on SVHN; and training the few-shot learner on MiniImageNet and testing on CUB-200.

**Network architectures:** Both the commentary and the student networks use a 4-block CNN architecture commonly seen in prior work (Finn et al., 2017). The student network takes a given image as input. The commentary network takes as input the support set image and the class label. The one-hot labels are converted into a 64 dimensional vector using a fully connected layer, concatenated with input image representations, then passed through two more fully connected layers to produce the output. This output is passed through a sigmoid to ensure example weights lie in the range $[0, 1]$. These weights are normalised to ensure a mean weight of 1 across the entire support set, which helped stability.

**Training details:** We use Adam with a learning rate of 1e-3 to learn both the commentary network parameters and the student network initialisation point for MAML. A meta-batch size of 4 is used for meta training. We use SGD with a learning rate of 1e-2 for the inner loop adaptation. At meta-training time, we use 5 inner loop updates. At meta-test time, we use 15 inner loop updates (similar to some other methods, to allow for more finetuning). For evaluation, we create 1000 different test time tasks (randomly generated) and we compute mean/95% CI accuracy on this set of tasks. We use the `higher` library (Grefenstette et al., 2019).

**Results:** We evaluate a standard MAML baseline and our commentary variant on standard few-shot learning benchmarks: (i) training/testing on MiniImageNet (MIN) and CIFAR-FS (*in-distribution testing*); and (ii) training on MIN and CIFAR-FS and testing on CUB-200-2011 (CUB), and SVHN (*out-of-distribution testing*). Results are shown in Table B.1. Each row specifies the experimental setting ($N$-way $K$-shot), the dataset used for training, and the dataset used for testing. In all experiments, incorporating example weighting can improve on the MAML baseline, suggesting the utility of these commentaries in few-shot learning.

| Mode | MAML Baseline | Example weighting + MAML |
|---|---|---|
| 5-way 1-shot: CIFAR-FS→CIFAR-FS | $46.27 \pm 0.71$ | $47.07 \pm 0.72$ |
| 5-way 1-shot: CIFAR-FS→SVHN | $24.89 \pm 0.36$ | $26.26 \pm 0.38$ |
| 5-way 5-shot: CIFAR-FS→CIFAR-FS | $65.40 \pm 0.61$ | $66.00 \pm 0.63$ |
| 5-way 5-shot: CIFAR-FS→SVHN | $37.28 \pm 0.43$ | $37.70 \pm 0.43$ |
| 5-way 1-shot: MIN→MIN | $43.64 \pm 0.61$ | $45.07 \pm 0.61$ |
| 5-way 1-shot: MIN→CUB | $37.45 \pm 0.55$ | $38.00 \pm 0.54$ |
| 5-way 5-shot: MIN→MIN | $61.85 \pm 0.52$ | $62.99 \pm 0.52$ |
| 5-way 5-shot: MIN→CUB | $57.75 \pm 0.54$ | $59.06 \pm 0.55$ |

**Table B.1:** Mean accuracy and 95% confidence interval across 1000 test-time tasks. On common few-shot learning benchmarks, using example weighting for the support set improves the MAML baseline's accuracy. Improvements are also observed when evaluating on out-of-distribution datasets.

## C  DATA AUGMENTATION

We provide further details about the experiments using commentaries to define data augmentation policies.

### C.1  MNIST

**Network and training details:** The 2 block CNN is used as the student network. Denoting each entry of the commentary grid as $\phi_{i,j}$, we initialised each entry to 0. The blending proportion is formed as: $\lambda_{i,j} = 1 - 0.5 \times \text{sigmoid}(\phi_{i,j})$. This is to ensure that the blending proportion is between 0.5 and 1; this implies that blended image contains more of the first image (class $i$) than the second (class $j$). Without this restriction, certain blending combinations could 'flip', making the results harder to interpret. The inner optimisation uses SGD with a learning rate of 1e-3, and had 500 gradient steps. We used 50 outer optimisation steps to learn the commentary parameters, using Adam with a learning rate of 1e-1. The commentary parameters were learned with the `higher` library (Grefenstette et al., 2019).

**Further Detail on MNIST Augmentation Commentaries:** We learn an augmentation commentary model $t$ on MNIST, represented as a $10 \times 10$ matrix. This commentary is learned by backpropagating through the inner optimisation, using a 2-block CNN student network. For each outer optimisation update, we use 500 steps of inner loop training with the augmented dataset, and then compute the validation loss on the unblended data to obtain commentary parameter gradients.

We find a trend in the learned augmentation relating the error rates and blending proportions. Consider a single image $x_i$, label $i$, and other images $x_j$, label $\forall j \neq i$. Averaging the blending proportions (computed as $0.5 \times \text{sigmoid}(\phi_{i,j})$) over $j$, the error rate on class $i$ is correlated (Pearson correlation$= -0.54$) with the degree of blending of other digits into an example of class $i$; that is, lower error rate on class $i$ implies that other digits are blended more heavily into it. On MNIST, this means that the class that has on average the lowest error rate (class 1) has other digits blended into it most significantly (seen in Figure 4 left). On the other hand, classes that have on average higher error rate (e.g., class 7, class 8) rarely have other digits blended in (Figure 4 right).

### C.2  CIFAR 10/100

**Network and training details:** The student network is a ResNet18. We use the method from Lorraine et al. (2020) to learn the commentary parameters. The commentary parameters are initialised in the same way as for the MNIST experiments. These parameters are learned jointly with a student, and we alternate updates to the commentary parameters and the student parameters. We use 1 Neumann step to approximate the inverse Hessian when using the IFT. For commentary learning, we use Adam with a LR of 1e-3 as the inner optimiser, and Adam with a LR of 1e-2 as the outer optimiser.

For evaluation, we train three randomly initialised students using the fixed commentary parameters. This training uses SGD with common settings for CIFAR (starting LR 1e-1, weight decay of 5e-4,

|  | CIFAR10 | | CIFAR100 | |
|---|---|---|---|---|
|  | Test Accuracy | Test Loss | Test Accuracy | Test Loss |
| No commentary | $93.84 \pm 0.22$ | $0.239 \pm 0.005$ | $74.79 \pm 0.39$ | $1.05 \pm 0.03$ |
| Random commentary | $93.84 \pm 0.30$ | $0.385 \pm 0.007$ | $73.39 \pm 0.58$ | $1.14 \pm 0.01$ |
| *mixup* (Zhang et al., 2018) | $94.42 \pm 0.12$ | $0.31 \pm 0.01$ | $75.89 \pm 0.55$ | $1.01 \pm 0.02$ |
| Learned commentary (shuffled) | $94.01 \pm 0.19$ | $0.235 \pm 0.005$ | $75.95 \pm 0.27$ | $0.99 \pm 0.01$ |
| Learned commentary (original) | $94.12 \pm 0.19$ | $0.225 \pm 0.007$ | $76.25 \pm 0.11$ | $0.97 \pm 0.01$ |

**Table C.1: Learned augmentation commentaries result in competitive or improved model accuracy and loss on CIFAR10 and 100.** Compared to not using a commentary and using a randomised label-dependent commentary, the original learned commentary results in models that are competitive with mixup and improve compared to other baselines. We show mean/standard deviation over three different initialisations.

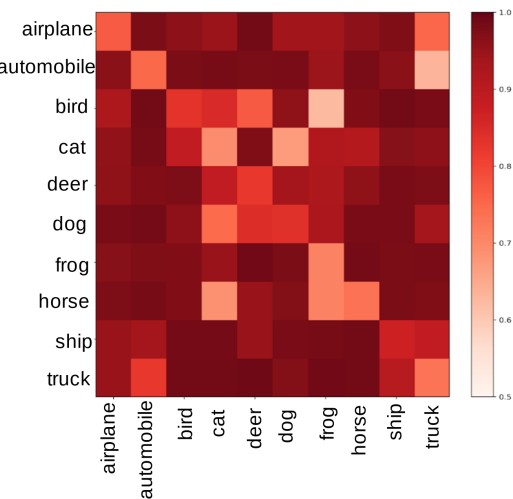

**Figure C.1:** Augmentation commentary policy on CIFAR10. We observe interesting emergent patterns of the learned blending proportions; classes that are visually similar and potentially confused (cat/dog, automobile/truck) are blended relatively significantly.

decaying LR after 30, 60, and 80 epochs by a factor of 10). We use standard CIFAR augmentations in addition to the learned augmentations at this evaluation phase.

**Baselines:** We compare to using no commentary (just the standard CIFAR augmentation policy, random crops and flips), a random commentary (where an augmentation grid is constructed by uniformly sampling blending proportions in the range $[0.5, 1]$), *mixup* (Zhang et al., 2018) with blending proportion drawn from Beta(1,1), and a shuffled version of our method where the commentary grid is shuffled at the start of evaluation (destroying the structure, but preserving the scale of augmentation).

**Results:** Table C.1 shows model accuracy for different augmentation policies on CIFAR10 and 100. We compare the learned commentary to using only standard data augmentations for CIFAR10/100 (No commentary), shuffling the learned commentary grid, using a random initialisation for the commentary tensor, and *mixup* (Zhang et al., 2018). We observe that the learned commentary is competitive with mixup and improves on other baselines. The fact that the learned commentary does better than the shuffled grid implies that the structure of the grid is also important, not just the scale of augmentations learned.

**Visualising the policy:** For CIFAR10, we visualize the full augmentation policy in the form of a blending grid, shown in Figure C.1. Each entry represents how much those two classes are blended, with scale on left. This corresponds to $0.5 \times \text{sigmoid}(\phi_{i,j})$, with $\phi_{i,j}$ representing an entry in the commentary grid.

|  | CIFAR10 | | CIFAR100 | |
|---|---|---|---|---|
|  | Test Accuracy | Test Loss | Test Accuracy | Test Loss |
| Random mask | $92.15 \pm 0.25$ | $0.274 \pm 0.008$ | $71.27 \pm 0.10$ | $1.18 \pm 0.02$ |
| Permuted learned mask | $91.69 \pm 0.07$ | $0.340 \pm 0.08$ | $71.50 \pm 0.07$ | $1.22 \pm 0.01$ |
| Learned mask | $92.99 \pm 0.11$ | $0.265 \pm 0.003$ | $73.16 \pm 0.44$ | $1.12 \pm 0.01$ |

**Table D.1:** Performance of different masking strategies on CIFAR10 and 100. Using the appropriate per-image learned masks improves (in both loss and accuracy) on permuting the masks across the entire dataset and randomly selecting mask regions of the appropriate scale.

## D   ATTENTION MASKS

**Datasets and Mask Information:** We use a number of datasets to evaluate the learned masks, including: Coloured MNIST (a synthetic MNIST variant), CheXpert (a dataset of chest X-ray images from (Irvin et al., 2019)), CIFAR10/100, and CUB-200-2011. More details:

- The Coloured MNIST dataset is formed by randomly sampling two MNIST digits from the dataset, and choosing one to be red and one to be blue. The red digit determines the image label. The two digits are randomly placed on two different quadrants of a $56 \times 56$ grid. The standard deviation of the mask is set to be 15 pixels.

- The CheXpert dataset has large X-ray radiograph images. Each image is resized to be $320 \times 200$. The mask standard deviation is 50 pixels.

- CIFAR10/100 masks are set to be 15 pixels standard deviation.

- CUB-200-2011 images are resized to be $224 \times 224$, and the mask standard deviation is 50 pixels.

**Network architectures:** The student network was a ResNet18, and the commentary network was a U-Net (Ronneberger et al., 2015) with an output layer from KeypointNet (Suwajanakorn et al., 2018). This takes a probability mass function defined spatially, and the $(x, y)$ centre of the mask is computed as the mean in both spatial dimensions. Producing the mean in this manner significantly helped stability rather than regressing a real value.

**Training details:** We use the method from Lorraine et al. (2020) to learn the commentary parameters. These parameters are learned jointly with a student, and we alternate updates to the commentary parameters and the student parameters. We use 1 Neumann step to approximate the inverse Hessian when using the IFT. When the commentary network is learned, we use Adam with LR 1e-4 for both inner and outer optimisations. We found balancing this learning rate to be important in the resulting stability of optimisation and quality of learned masks.

When evaluating the learned masks, we trained three new ResNet-18 students with different random seeds, fixing the commentary network. For CIFAR10/100, for evaluation, we use SGD with common settings for CIFAR (starting LR 1e-1, weight decay of 5e-4, decaying LR after 30, 60, and 80 epochs by a factor of 10). We use standard CIFAR augmentations for this also.

**Visualizing Masks:** Figure D.1 shows masks from the main text and further additional examples.

**Baselines for CIFAR experiments:** For the random mask baseline, for each example in a batch, we select a centre point randomised over the whole image, then form the mask by considering a gaussian centered at that point with standard deviation 15 (same size as masks from commentary network). This resembles very aggressive random cropping. For the permuted learned mask, we use the learned commentary network to predict masks for all images. Then, we permute the mask-image pairs, so that they no longer match up. We then train with these permuted pairs and assess performance. Our goal is to understand what happens when we have random masks with a similar overall spatial distribution to the real masks.

**CIFAR Masking Quantitative Analysis:** We compare masks from the learned commentary network to two baselines: randomly chosen mask regions for each image (of the same scale as the learned masks, but with the centre point randomised over the input image), and permuted masks

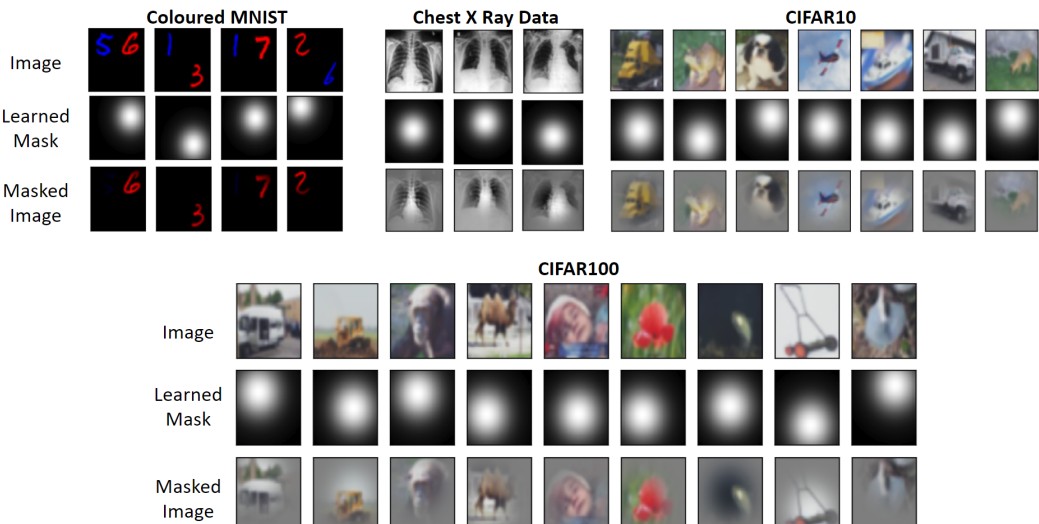

**Figure D.1:** Learned attention masks highlight salient image regions for classification. We learn a commentary network to produce image attention masks on four datasets, and the masks are qualitatively effective in all cases. On Coloured MNIST, where the image label is determined by the red digit (blue is a distractor), the masks focus on the red digit. On a dataset of chest X-rays, the masks focus on the chest cavity, which is the appropriate reason for detecting the condition in question (cardiomegaly). On CIFAR10/100, the masks are focused on important regions of the object in the image, such as the faces of animals/babies, and the bodies of vehicles.

(where we shuffle the learned mask across all the data points). Table D.1 shows the results. Especially on CIFAR100, the learned mask improves noticeably on the other masking methods in both test accuracy and loss. This suggests that overall, the masks are highlighting more informative image regions. We do not expect using the masks on standard problems to result in improved held-out performance, because the backgrounds of images may contain relevant information to the classification decision.

**Further details on robustness study:** The dataset was generated using the open source code from Koh et al. (2020). The student network for this study was pretrained on ImageNet, as in Koh et al. (2020). To train student models at the evaluation stage, we used SGD with a learning rate of 0.01, decayed by a factor of 10 after 50 and 90 epochs. We used nesterov momentum (0.9) and weight decay of 5e-5.

