# OpenReview forum: "Teaching with Commentaries"
_ICLR.cc/2021/Conference — ICLR 2021 Poster_

### Official Review · AnonReviewer3 · 2020-10-26
**Missing important details, inconsistent baselines**

**Rating:** 5
**Confidence:** 5

**Review:**

[Summary] This paper introduces a gradient-based meta-learning method to learn weighting for each training sample, called "commentary" in the paper's description, as a form of auxiliary learning, updated on training loss to accelerate training speed, and improve generalisation.

[Strength] The motivation is interesting, as to balance training by automatically computing training sample weighting. The toy example on rotated MNIST clearly shows that the teacher network would put a higher weighting on samples closing to the decision boundary, which is expected and confirms the authors' assumption.

[Weakness] The paper has several weaknesses which I will outline below.
-- Curriculum v.s. Non-curriculum learning. The authors present two versions of "commentary", one is with a curriculum and the other is without a curriculum. The curriculum version, which I assume is to update training sample weighting iteratively with the updated teacher network. However, the authors did not mention any details on the non-curriculum learning version, on how they compute the training sample weighting in a fixed way.

-- Inconsistent results on ResNet18. In Fig 3., ResNet 18 seems to perform unreasonably low on CIFAR-10/100 with roughly ~70 and ~45 of accuracy, respectively. However, with the same network of ResNet18, they perform in a reasonable performance in Table 2.

-- Marginal performance. Using commentary to generate mix-up style sample weighting seems reasonable and interesting. But the results are quite marginal, or do not consistently improve over the standard mix-up baseline as shown in Table 2.

Further questions: What is the distribution of training and validation data in CIFAR10/100 results? Is it possible that the teacher network simply put higher weights on training samples that look most similar to validation data, as a form of a trivial/degenerated solution? I assume this might happen particularly in the rotated MNIST example? And how authors generate the labels for those overlapping samples in rotated MNIST?

I hope the authors could elaborate on these questions and further refine the paper. I think in general the idea is interesting and worth exploring, but this paper is not ready yet.

---

> ### Author Response · Authors · 2020-11-18
> **Author response to AnonReviewer3**
>
> We thank the reviewer for their comments.
>
> **Curriculum learning:**  Curriculum learning in our framework is very straightforward to encode: the commentary network simply takes as additional input the student training step (along with the input image) and outputs a commentary. The commentary network is then learned using the methods outlined in Section 2. (Note we refer to this as a curriculum learning commentary because the commentary evolves over the course of student training, and can weight examples differently depending on the stage of training.)
>
> In contrast, the non-curriculum variant has a commentary network that only takes as input the image and outputs the per-example weight, with no dependence on the stage of training.
>
> We will clarify this distinction in a revision of the paper.
>
>
> **Performance of ResNet18:** We note that the ResNet18 performance is not inconsistent: in Figures 2 and 3, only the initial portion of training is shown (first few epochs) rather than showing performance at convergence. We train the network to convergence in Table 2, thus these numbers are different.
>
> This is because the goal of Figures 2 and 3 is to show that using commentaries with curriculum structure (detailed more above) can result in improvements in training speed during the early portion of training. Furthermore, although the commentary network was trained with a particular student network architecture for a small number of inner optimization steps, the commentary network’s weights can generalize to speeding up training for (i) much longer time horizons (Fig. 2) and different network architectures (Fig. 3).
>
>
> **Performance in data augmentation experiments:** We agree that accuracy is comparable to mixup, but note that our method does have consistent improvements in test loss (likely due to the fact that the method optimizes validation loss not accuracy directly.)
>
> More importantly, a key goal in the paper is to use commentaries to obtain insights about training networks, rather than only demonstrating performance improvements. Through analyzing our learned data augmentation commentaries on MNIST, CIFAR-10 and CIFAR-100, we are able to gain such insights on e.g. connections between learned data augmentation, class error rates and visual sample similarity.
>
>
> **Train/validation distributions:** We first note that we use held-out test sets (completely disjoint from training and validation sets) when reporting all results, so if such a degeneracy were to exist, it would likely manifest as worsened test set performance.
>
> Our training and validation sets are generated by randomly splitting the entire set of data available for training, and where applicable, we repeat over 3 random splits. These sets have similar distributions, and we have analyzed the resulting example weights and do not see the kind of degeneracy mentioned.
>
> On the Rotated MNIST domain, training and validation sets are obtained from the same generative distribution. For each index in the dataset, we uniformly sample a binary label. We then sample the rotation from the specified uniform distributions, conditioned on the label. We follow the same process for the overlapping and non-overlapping settings; the only change is the class-conditional rotation distributions.
>
> In terms of results: we provide detailed analysis for this domain, illustrating the close relationship between the rotation of the digit and the resulting example weighting (see Fig 1, and results in Appendix C). In particular, the evolution of the weights over student network training (the learned curriculum, see Appendix C) shows sensible progression, and no evidence of degenerate structure.

---

### Official Review · AnonReviewer1 · 2020-10-27
**A general framework to learn meta-information via properly designing 'commentary'**

**Rating:** 7
**Confidence:** 3

**Review:**

This paper proposes a general framework for boosting CNNs performance on different tasks by using'commentary' to learn meta-information. The obtained meta-information can also be used for other purposes such as the mask of objects within spurious background and the similarities among classes. The commentary module would be incorporated into standard networks and be iteratively optimized with the host via the proposed objective. To effectively optimize both the commentary and the standard network, this paper adopts the techniques including implicit function theorem and efficient inverse Hessian approximations.

This paper studies three kinds of commentary named weight curriculum commentary that learns individual weights for data samples, augmentation commentary that learns data augmentation strategy similar to mix-up, and attention mask commentary that learns to focus on objects of images. All three commentaries have been examined via extensive experiments on small-scale benchmarks and shown improvements.

The studied direction is very interesting that how to obtain meta-information with standard data annotations. Although, the reviewer thinks all three commentaries' tasks are studied in the literature. This paper integrates them into a general framework. Also, the weight curriculum commentary can be naturally compatible with MAML. There may be possible to derive more different auxiliary tasks via properly designing the commentary structure. The experimental details are described in detail and make the experiments are easy to replicate. Overall the paper is easy to follow.

-------------------------

However, the proposed framework needs human efforts to specifically design the formula of the commentary structure to involve proper inductive bias for learning the targeted meta-information. There're works that also only use the standard annotations to study adjusting the weights for data samples, data augmentation (like the mix-up), and focusing the object.  Compared to them, the reviewer does not see the clear advantages of the proposed method. Also, the author does not provide many comparisons.

The reviewer also concerns about scalability. Currently, the method is only tested on several small-scale benchmarks including MNIST, CIFAR, TINY-ImageNet, and CelebA, X Ray Data. However, the objective would need a lot of computation resources, though approximately computing the inverse Hessian. The reviewer doubts that it is hard to apply in real-world tasks and the method can not bring benefits when the data is big. But, this concern may be beyond the scope of this paper and can be considered in the following works.

In conclusion, the reviewer would rate 7 due to it nicely integrates several meta-information tasks into a unified framework and hope this work can bring some new understandings to the communities.

---------------after rebuttal----------

I've read all reviews and the rebuttal. I think overall this is a good paper and would like to keep my score.

---

> ### Author Response · Authors · 2020-11-18
> **Author response to AnonReviewer1**
>
> We thank the reviewer for their comments.
>
> **Human effort to identify commentary structure:** We agree that some human effort is required to specify the structure of the commentary. However, we emphasize that there already exists a large number of such structures/inductive biases that could be naturally incorporated in this framework. We explore example weighting, blending-based data augmentation, and Gaussian attention masks in most detail in this work, but there are many other useful structures that can be framed as commentary learning, including:
> * Multitask learning/auxiliary targets: we explore this briefly in appendix B for CelebA, and other work also considers this setting ([1])
> * Other forms of data augmentation: in [2], the authors explore various other forms of data augmentation (rotations, cropping, shifting, etc).
> * Loss function structure [3]
>
>
> **Comparisons:** Many prior works are not directly comparable to our more general framing since they design algorithms specific to a particular setting (for example, auxiliary task learning [1]). We do provide comparisons to baselines and ablations in our quantitative experiments: for example weighting, an ablation without curriculum structure when assessing learning speedups;  a MAML baseline when examining few-shot learning performance; data augmentation methods such as mixup, and ablations that incorporate random blending structure or destroy the learned blending structure; and random masking/shuffled masking baselines when evaluating metalearned attention masks commentaries.
>
> **Scalability:** This approach is readily scalable due to the efficient inverse Hessian approximation introduced by [4] (similar time complexity to a single training iteration). In our experiments, we also evaluate on larger scale datasets -- the Chest X-ray datasets is large and consists of images that are 320 x 200 pixels, and CUB is a 200 class problem with 224 x 224 images. We agree that there are even larger scale experiments to try (e.g. ImageNet scale) but recent work ([2]) has applied a similar implicit differentiation approach on ImageNet, demonstrating that the method can indeed be scaled to such tasks.
>
> [1] Self-Supervised Generalisation with Meta Auxiliary Learning, Liu et al.
>
> [2] Meta Approach to Data Augmentation Optimization, Hataya et al.
>
> [3] Learning to Teach with Dynamic Loss Functions, Wu et al.
>
> [4] Optimizing Millions of Hyperparameters by Implicit Differentiation, Lorraine et al.

---

> > ### Comment · AnonReviewer1 · 2020-11-24
> > **Response**
> >
> > Thanks for your reply. I encourage you to add such a discussion of limitations in your paper. For scalability, I understand the limitation of time and computing resources. But the scale of CUB and the task of the Chest X-ray are not suitable to prove your method is general. If there's possible, I encourage you to test a bit and add results in the revision.

---

### Official Review · AnonReviewer2 · 2020-10-28
**An interesting Meta-Learning Approach**

**Rating:** 7
**Confidence:** 4

**Review:**

The authors propose a framework for learning a commentary or teacher model
that provides helpful information during training. Under this framework, a
teacher model learns to provide meta-information, referred to as a commentary,
on a given training example, with the goal of improving the student network
(e.g., improving convergence speed or robustness). The learning of the teacher
model is done by minimizing the validation set loss of the student model under
parameterization from a training loop that used the commentaries.

The optimization of the teacher parameters involves backpropagating
through gradient descent steps. The authors propose doing this directly
in small toy examples, or using an approximation using the Implicit Function
Theorem for more realistic problems.

The authors then demonstrate that the commentary learning framework can
be used in a variety of ways:
learning to provide example weights, learning a blending policy for data augmentation, and learning to provide an attention mask for image classification.
In the latter case, they show that using the attention masks leads to a more
robust student when the backgrounds are modified to spuriously correlate
particular class labels on the training/validation sets but not on the
test set.

This paper has many strengths. It introduces an interesting meta-learning
framework and demonstrate its flexibility to implement three different
kinds of commentary for improving a student network. Additionally, it supports
its statements with a variety of interesting experiments. While I would
like to see this paper accepted, it could be improved
in its organization and explication.

For example, it is not immediately clear that the commentary network is learned with equations 1 & 2, but the student network in this process is ultimately discarded and only after the finished teacher network is obtained, is a
student network actually trained for evaluation purposes. An "illustrative example" of commentaries is put in the appendix instead of the main paper which
feels like a lost opportunity to holistically explain the algorithm and
clear up these misconceptions.

In the example weighting curricula it is not clear what is meant by curricula.
In an ablation study, example weights produced by a teacher network with and
without curricula are compared, but what constitutes the curricula here is
never defined. I'm inferring that in the curricula case, the teacher network
produces weights using both the example x and the current training iteration
i, while the non-curricula case only uses x. This should be stated explicitly
somewhere.

In equations 1 and 2 the authors write

optimize(phi/theta)[loss func]

when

argmin_phi/theta loss func

would convey what the authors intend while using standard notation.

On the masking experiments, the masks are used both at training and test time
which seems somewhat odd. The commentary formalism proposed is supposed to
be limited to learning a commentary model to facilitate training. In this case,
the commentary model seems less like a separate teaching model and more like
another component of the classification model, similar to "Recurrent Models of Visual Attention" where a component of the classifier determines where to
focus in an image. How well does the student network trained with masks provided by the teacher do on the validation/test sets without masks?

---

> ### Author Response · Authors · 2020-11-18
> **Author response to AnonReviewer2**
>
> We thank the reviewer for their comments.
>
>
> **Regarding discarding the student:** Note that we do *not* have to discard the student and train a new one -- it is possible to directly utilize this student network for evaluation.
>
> However, in our experiments, we were interested in seeing whether the learned commentary could generalize to new student networks with different random initializations, which is why we trained a new student with the (fixed) learned commentary as evaluation. For the example weighting section in particular, we wanted to explore generalization across longer time horizons and across different architectures, hence we chose this approach. We will clarify this motivation in an updated version of the paper.
>
> **Curriculum learning:** This is indeed what we mean when we say curriculum learning; we thank the reviewer for this comment and will update the paper to be clearer.
>
>
> **Usage of optimize vs argmin:** We chose to use optimize since we do not train the student network to convergence at each step of training the teacher commentary network. However, we acknowledge this may be unclear so will revise it in an updated version.
>
>
> **Usage of masks for validation/test data:** We view the learned masks as a *metalearned preprocessing strategy* to improve robustness and provide interpretable value. As a result of this view, we opt to mask the validation/test images also.  In future work we hope to further explore related questions, with different types of masks and masking strategies.

---

### Official Review · AnonReviewer4 · 2020-10-28
**Interesting concept that ties together disparate ways of using meta-information for network training, concerns around scalability**

**Rating:** 6
**Confidence:** 4

**Review:**

Summary: This paper proposes a framework for incorporating and optimising meta-information as additional parameters when optimising the validation loss for a specific network on a given task. More precisely, they instantiate a teacher network that supplies the meta-information (could be anything from attention maps, to importance weights over sample points, to parameters of a data-augmentation scheme) that is then used by a student network, and propose a formulation whereby the two are trained in tandem so as to optimise the output loss of the student network.

General remarks: The paper is well-written and well-presented. The idea is clear, language is effective and visualisations/tabulations are accessible.

Pros (Originality/Significance)
(a) I believe the biggest contribution of the paper is in the unifying of different existing approaches that explicate and optimise meta-information as part of the network pipeline in a bid to improve performance (accuracy or robustness). Ideas such as example reweighting, example blending (e.g. mixup), describing attention masks, or transforming input data (rotation, etc) are treated as special cases of a particular student-teacher formulation. To the best of my knowledge, this unified treatment is unique and comes at a crucial time when same techniques seem to reappear under different names and in different forms and prevent a wholecome understanding of the approach and its wider implications.

Cons (Originality/Significance)
(a) The maths of the reunification is rather simple, teacher outputs values that a student uses and optimises, and could lead to a time-consuming training.
(b) The instantiations of meta-information studied in this paper have been explored before and don't seem to offer any significant advantages over existing methods (the benchmarks in some tables are not up-to-date, for example, consider referring to [1] for sample reweighting benchmark and [2] wherein an attention based classification network for CUBS has been proposed.
(c) Some use cases are shown to work only for small datasets, such as CIFARs for example reweighting. Newer techniques that are able to scale example reweighting to ImageNet exist [3]. Their method seems to need curriculum learning on top of example reweighting for full benefit, please comment.


[1] Learning to Reweight Examples for Robust Deep Learning, Mengye Ren et.al.
[2] Learn to pay attention, Jetley et.al.
[3] Sample Balancing for Deep Learning-Based Visual Recognition, Chen et.al.

---

> ### Author Response · Authors · 2020-11-18
> **Author response to AnonReviewer4**
>
> We thank the reviewer for their comments.
>
> **With regards to time consuming training:**  although this may be the case with using backpropagation through training, using the implicit differentiation method, Section 2.1 Eq 3 (as we do for all our larger scale evaluations) has similar time complexity to training normally (see [1]). Thus, our method does not necessarily require time consuming training.
>
> **Instantiations of meta-information:** Our main goals in this paper were to: (i) formalize our unifying commentaries framework with associated learning algorithms; (ii) explore potential applications; and (iii) demonstrate performance improvements and insights on learning that could be obtained with the learned commentaries.
> To this end, we explored three representative instantiations: example weighting, blending examples for data augmentation, and learning structured attention masks to identify salient image regions. We conduct quantitative evaluation on a range of datasets including natural image classification (CIFAR10/100), face attribute classification (CelebA), fine-grained classification (CUB), medical images (Chest X-rays), and few-shot learning (MiniImageNet, CUB, CIFARFS, SVHN).  Using our framework in a diverse range of settings illustrates its flexibility.
>
> Although certain instantiations we present may exist in prior work (e.g. example weighting), we evaluate new applications of these instantiations (for example, using example weighting to improve the MAML algorithm for few shot learning, and metalearning blending-based data augmentation strategies). The flexibility of our framework makes it easy to explore these new applications.
> We thank the reviewer for the additional helpful references, but we note that some of these do not quite fit into our framework; for example, the paper by Jetley et al. mentioned learns attention maps in an end-to-end fashion rather than using metalearning. These masks are also more complex in structure to those that we examine; in future work we hope to further explore related strategies, with different types of masks and masking strategies.
>
>
> **Scaling up applications of commentaries:** We note that with the implicit differentiation method, commentary learning can be applied in larger scale settings. Indeed, both Chest X Ray and CUB classification tasks have larger sized images: the chest X ray images are 320 x 200 dimensional (the dataset has 224,316 images overall), and CUB images are 224 x 224 (the dataset has 11,788 images). In addition, there is interesting related work using implicit differentiation to metalearn data augmentation at ImageNet scale ([2])
>
>
> **Clarification on curriculum learning for example weighting:** Curriculum learning in our framework is very straightforward to encode: the commentary network simply takes as additional input the student training step (along with the input image) and outputs an example weight. The commentary network is then learned using the methods outlined in Section 2. (Note we refer to this as a curriculum learning commentary because the commentary evolves over the course of student training, and can weight examples differently depending on the stage of training.)
>
> In contrast, the non-curriculum variant has a commentary network that only takes as input the image and outputs the per-example weight, with no dependence on the stage of training.
>
> The non-curriculum variant in Figures 2 and 3 represent an ablation study -- we evaluate this to illustrate how the curriculum structure (taking iteration of training as input) is important to achieve learning speedups. We provide further intuition on the benefit of curriculum structure in Appendix C.
>
> [1] Optimizing Millions of Hyperparameters by Implicit Differentiation, Lorraine et al.
>
> [2] Meta Approach to Data Augmentation Optimization, Hataya et al.

---

### Public Comment · ~Jordan_FRERy_y1 · 2020-11-13
**Student + Comentary networks vs Student trained on train + validation comparison**

The auteurs use a commentary network that uses a validation to get gradient that will be propagated to the student network.

This approach feels like the commentary network could use the validation to provide validation set information to the student that are missing from the training.

I would be very interested to know whether the commentary network provides more information to the student compared to a student network that would have access to the validation set.

---

> ### Author Response · Authors · 2020-11-18
> **Response to Public Comment**
>
> Thanks for your interest and question! We believe that the possibility of information leakage from the validation set is low in our experiments, given that the (per example) output commentaries are relatively low capacity -- e.g. a scalar/2D vector per example -- which makes it difficult to leak information. They are also incorporated in a very specific way into the loss function, further reducing the chance of leakage. We also conducted experiments using the validation set in training for the example weighting and masking, and found that the reported trends still held.

---

### Decision · Program_Chairs · 2021-01-07
**Final Decision**

**Decision:**

Accept (Poster)

**Comment:**

This paper proposes an interesting unified framework for meta-learning with commentaries, which contains information helpful for learning about new tasks or new data points. The authors present three kinds of different instantiations, i.e., example weighting, example blending, and attention mask, and show the effectiveness with the extensive experiments. The proposed method has a potential to be used for a wide variety of tasks.